# Radioprotective Effects of Kelulut Honey in Zebrafish Model

**DOI:** 10.3390/molecules26061557

**Published:** 2021-03-12

**Authors:** Mohd Noor Hidayat Adenan, Latifah Saiful Yazan, Annie Christianus, Nur Fariesha Md Hashim, Suzita Mohd Noor, Shuhaimi Shamsudin, Farah Jehan Ahmad Bahri, Khairuddin Abdul Rahim

**Affiliations:** 1Agrotechnology and Biosciences Division, Malaysian Nuclear Agency, Bangi, Kajang 43000, Malaysia; hidayat@nm.gov.my (M.N.H.A.); shuhaimi_s@nm.gov.my (S.S.); kabdulrahim57@gmail.com (K.A.R.); 2Institute of Bioscience, Universiti Putra Malaysia, Serdang 43400, Malaysia; annie@upm.edu.my (A.C.); farahjehan.bahri@yahoo.com (F.J.A.B.); 3Department of Biomedical Science, Faculty of Medicine and Health Sciences, Universiti Putra Malaysia, Serdang 43400, Malaysia; nurfariesha@upm.edu.my; 4Department of Biomedical Science, Faculty of Medicine, University of Malaya, Jalan Universiti, Kuala Lumpur 50603, Malaysia; suzita@um.edu.my

**Keywords:** radioprotection, Kelulut honey, *Trigona* sp., zebrafish (*Danio rerio*) embryos

## Abstract

Large doses of ionizing radiation can damage human tissues. Therefore, there is a need to investigate the radiation effects as well as identify effective and non-toxic radioprotectors. This study evaluated the radioprotective effects of Kelulut honey (KH) from stingless bee (*Trigona* sp.) on zebrafish (*Danio rerio*) embryos. Viable zebrafish embryos at 24 hpf were dechorionated and divided into four groups, namely untreated and non-irradiated, untreated and irradiated, KH pre-treatment and amifostine pre-treatment. The embryos were first treated with KH (8 mg/mL) or amifostine (4 mM) before irradiation at doses of 11 Gy to 20 Gy using gamma ray source, caesium-137 (^137^Cs). Lethality and abnormality analysis were performed on all of the embryos in the study. Immunohistochemistry assay was also performed using selected proteins, namely γ-H2AX and caspase-3, to investigate DNA damages and incidences of apoptosis. KH was found to reduce coagulation effects at up to 20 Gy in the lethality analysis. The embryos developed combinations of abnormality, namely microphthalmia (M), body curvature and microphthalmia (BM), body curvature with microphthalmia and microcephaly (BMC), microphthalmia and pericardial oedema (MO), pericardial oedema (O), microphthalmia with microcephaly and pericardial oedema (MCO) and all of the abnormalities (AA). There were more abnormalities developed from 24 to 72 h (h) post-irradiation in all groups. At 96 h post-irradiation, KH was identified to reduce body curvature effect in the irradiated embryos (up to 16 Gy). γ-H2AX and caspase-3 intensities in the embryos pre-treated with KH were also found to be lower than the untreated group at gamma irradiation doses of 11 Gy to 20 Gy and 11 Gy to 19 Gy, respectively. KH was proven to increase the survival rate of zebrafish embryos and exhibited protection against organ-specific abnormality. KH was also found to possess cellular protective mechanism by reducing DNA damage and apoptosis proteins expression.

## 1. Introduction

Ionizing radiation is widely used in various fields of medicine, industry and agriculture. Radioisotopes are used extensively in diagnosis and therapy for several diseases. For medical diagnostic imaging, X-rays can provide images to identify abnormal changes in body and tissues. The radioactive isotopes from gamma-ray sources have been used in cancer treatments by destroying abnormal forms of cells [1]. In industrial sectors, X-rays and gamma rays are used to generate images of the interior of solid materials, as a means of nondestructive testing (NDT) and inspection. NDT radiography is used in the petroleum, chemical and nuclear industries to test consumer goods. For example, pipelines are inspected both during installation and maintenance to ensure that welds remain intact [2]. Ionizing radiations are employed in agricultural field as a research tool to develop new varieties of agricultural crops with desired attributes, for example drought and disease resistant, high quality, short growing period and high yielding. Previously, a mutation breeding technique was performed by utilizing gamma radiation from caesium-137 (^37^Cs) radioisotope source to develop new kenaf plant variants or mutants that are adaptable to a wide range of conditions and environmental factors [3].

As the use of ionizing radiation increases, so does the risk for health hazards if not properly used or contained. Low doses of ionizing radiation can increase the risk of longer-term effects such as cancer [4]. Radiation overexposure accidents are possible to cause acute health effects such as skin burns or acute radiation syndrome can occur when doses of radiation exceed certain levels [5]. From year 1980 to 2013, 634 radiation accidents were reported, encompassing 2390 overexposed people, of whom 8% died from the overexposure. Among the reported radiation accidents, most of them occurred in medical and industrial sectors. Number of overexposed people in medical sector was the highest (64%) followed by accidents in industrial sector (22%), and orphan sources (9%). The number of deaths resulting from radiation accidents reported in radiation therapy was the greatest (51%), followed by those reported in the industrial sector (24%) and accidents involving orphan sources (19%) [6].

Exposure of radiation leads to the formation of free radicals, which induces biological damage even at a very low dose [7]. Damage due to ionizing radiation exposures contributed to DNA strand breaks either single or double strand breaks [8]. In response to the strand breaks formation, subtype of histone H2A called H2AX is phosphorylated to γH2AX. H2AX phosphorylation plays an important role in DNA damage response [9]. Following the induction of DNA damage, a leading route of cell inactivation is apoptosis [10]. Apoptosis is a secondary response to DNA damage in order to protect a multicellular organism against a damaged cell [11]. Ionizing radiation exposure often triggers the onset of p53-dependent apoptotic pathways or known as intrinsic apoptotic pathways [12]. This involves activations of proteins such as puma, bax, cytochrome C, Apaf-1, caspase-9 and caspase-3. Ionizing radiation also can cause bystander effects which is a phenomenon that non-irradiated cells exhibit effects along with their different levels as a result of signals received from nearby irradiated cells [13]. These effects are critically dependent on intercellular communication between the irradiated cells and bystanders [14]. 

Therefore, there is still a great need to investigate and understand the effects of ionizing radiation on human and find suitable radiation protection agents, or radioprotective agents with capability of alleviating adverse effects of radiation exposure. There is also a need to develop an effective and non-toxic radioprotector. The currently available ones have many drawbacks including side effects and toxicity. The Walter Reed Army Institute has developed a radioprotector called amifostine to protect personnel from radiation sickness [15]. However, clinical application of amifostine is currently limited due to its harmful side effects, including nausea and vomiting [16].

Kelulut or stingless bees are the oldest bees discovered; there are about 500 Kelulut species around the world that can be found in several tropical areas of Australia, Africa, America and in Southeast Asian countries including Malaysia. Compared to other Asian tropical countries Malaysia, with more than 35 species, is reported to have the highest diversity of *Trigona* species. Normally, Kelulut bees develop their nests on roots or chopped down tree wood [17]. KH is suitable to develop as natural radioprotector because it displays higher medicinal values as compared with other honey products. It was recorded in previous study that the total phenolic acid content of honey from stingless bee was higher than those from *Apis* spp [18]. In addition, a previous investigation reported that stingless bee honey which displayed a high phenolic content has a slightly higher ABTS+ cation radical scavenging capacity [19]. This result indicates a correlation between the phenolic content and antioxidant activity of stingless bee honey [20]. Phenolic acid derivatives can reduce inflammation, the proliferation of cancer cells and the activity of pathogenic microorganisms [18]. 

The use of a strong model organism that has higher compatibility with human is required for any radiobiological studies. The zebrafish (*Danio rerio*) was selected in the present study since this organism’s genome possesses approximately 70% similarity with human genes [21], and thus can be a suitable model organism. Zebrafish is a freshwater aquarium fish that has become a powerful and popular model organism for research into vertebrate genetics, development and toxicology during the last 40 years. Several advantages of this model have led to its rapid growth as a research model including their ease of care, fecundity, rapid development, small size and ease of manipulation. An extensive sharing of techniques, reagents, and fish lines within the zebrafish research community has facilitated the rapid growth of research using zebrafish, with a tremendous increase in the number of publications using the fish in recent years [22]. Additionally, the uniqueness of this fish which have transparent embryos when using specially formulated media is another advantage to evaluate or investigate their developmental growth based on observation of their basic organs. The organs such as eyes, heart, spine and brain are always used in developmental toxicity analysis studies. This particular study was conducted to determine the effects of KH treatment on irradiated and bystander zebrafish embryos. 

## 2. Results

### 2.1. Lethality Effects in the Zebrafish Embryos

Figure 1 depicts normal morphology/development and lethality (coagulation) in zebrafish embryos from the treatment groups. The coagulation effect only occurred at 48 h post fertilization (hpf) and 72 hpf.

The coagulation in zebrafish embryos from the irradiated without treatment (IN), irradiated with KH treatment (IKH) and irradiated with amifostine treatment (IA) groups was observed at 24 and 48 h post-irradiation. The percentage of coagulated zebrafish embryos at 24 and 48 h post-irradiation is shown in Figure 2. Based on the data, 40% to 75% coagulated zebrafish embryos were noted in the IN group at gamma irradiation doses of 14 Gy to 20 Gy and no coagulated zebrafish embryos were observed in the IA group at 24 h post-irradiation (Figure 2a). Coagulated embryos in the IKH and IA groups were also found to be lower than IN group at 24 h post-irradiation with a percentage range of 0% to 40% at 14 Gy to 20 Gy (Figure 2a). At 48 h post-treatment, 12% to 27% coagulated zebrafish embryos were identified in the IN and IKH groups at 17 Gy to 20 Gy of irradiation dose (Figure 2b). There were no coagulated embryos noted at 17 and 18 Gy of irradiation dose in the IA group at 48 h post-irradiation (Figure 2b). 

The total percentages of coagulated embryos from 24 h to 48 h post-irradiation for each irradiation dose in IN, IKH and IA group are shown in Figure 3. The highest number of coagulated embryos were noted at 14 Gy to 20 Gy of irradiation dose in the IN group with percentages of 40% to 92%. Coagulation incidence in the IA group was significantly lower at 17 Gy, 18 Gy and 20 Gy of irradiation dose with percentages of 0% to 32%, respectively. Total coagulated percentages of embryos in the IKH group were found to be significantly lower than IN at 14 Gy to 20 Gy (0% to 67%, respectively). Lethality effects in the IKH and IA groups are found to be similar at 14 Gy, 15 Gy, 16 Gy and 19 Gy.

### 2.2. Abnormality Effects in Zebrafish Embryos

Normal development and abnormalities of zebrafish embryos from all of the treatment groups at 24 h to 96 h post-irradiation (48 hpf to 120 hpf) are shown in Figure 4. Three types of abnormality were found in the embryos at 24 h post-irradiation: microphthalmia, body curvature with microphthalmia and body curvature with microphthalmia and microcephaly (Figure 4a). Microphthalmia and body curvature with microphthalmia were found in the IN group and labelled as M (IN) and BM (IN), respectively. Other than that, microphthalmia was also identified in the IKH group and labelled as M (IKH) while body curvature with microphthalmia and microcephaly was developed in the IA group and assigned as BMC (IA). 

The development of zebrafish embryos was observed continuously. At 48 h post-irradiation (72 hpf), the abnormalities found were microphthalmia and pericardial oedema (MO), body curvature and microphthalmia (BM) and body curvature with microphthalmia and microcephaly (BMC). MO was identified in the IN and IKH groups and assigned as MO (IN) and MO (IKH), respectively. Furthermore, BM (IN) and BMC (IA) perpetually occurred at 48 h post-irradiation. As the time increased, abnormalities could also be found in the zebrafish embryos irradiated with 13 Gy. Figure 4b shows the abnormalities developed in the zebrafish embryos at this growth stage.

Abnormalities of the embryos at 72 h post-irradiation (96 hpf) are depicted in Figure 4c. Based on the results, MO (IN) still occurred in the zebrafish embryos. At this stage, microphthalmia, microcephaly, body curvature and pericardial oedema (AA) was identified in all of the treatment groups and assigned as AA (IN), AA (IKH) and AA (IA). Several zebrafish embryos from the IKH group developed only pericardial oedema (O) abnormality at this growth stage (O (IKH)).

The final stage of observation was conducted at 96 h post-irradiation (120 hpf). At this time, only AA (IN) and AA (IKH) can be found in zebrafish embryos. Other abnormalities identified were microphthalmia with microcephaly and pericardial oedema (MCO) in IKH and IA group. The developed abnormalities at 96 h post-irradiation are depicted in Figure 4d.

Figure 5a shows the percentage of abnormalities in each treatment group at 24 h post-fertilization. M (IN) was noted as the highest abnormality at 14 Gy and 15 Gy as compared to other abnormalities with percentages of 32 and 25%, respectively. There was no M (IN) noted at 17 Gy to 20 Gy of irradiation dose. Based on the data, BM (IN) was identified in zebrafish embryos irradiated at 16 Gy to 20 Gy but only noted at 16 Gy and 20 Gy with the percentages of 28 and 25%, respectively. On the other hand, M (IKH) was found in embryos that has been irradiated at 17 Gy to 20 Gy and only noted at 18 Gy with the percentage of 45%. BMC (IA) was found in embryos irradiated at 17 Gy to 20 Gy but 15% of it was noted at 17 Gy. Percentage of BMC (IA) increased with increasing of irradiation dose. Abnormalities in IKH and IA groups were found to be lower than IN group at 14 Gy to 16 Gy and 14 Gy to 17 Gy respectively at 24 h post-irradiation. 

Percentages of abnormalities in zebrafish embryos from each treatment group at 48 h post-fertilization are shown in Figure 5b. After irradiation with doses of 13 Gy to 20 Gy, MO (IKH) abnormality was developed in the embryos but only noted at 13 Gy, 15 Gy, 16 Gy and 20 Gy with the percentages of 22%, 40%, 68% and 33%, respectively. MO (IN) was not significantly recorded at 14 Gy and 15 Gy with the percentage of 32% and 25%, respectively. No MO (IN) was observed at 16 Gy, 18 Gy and 19 Gy as compared to other abnormalities. BM (IN) developed in the embryos after being irradiated with 16 Gy to 20 Gy and only observed at 18 Gy with the percentage of 18%. No BM (IN) was observed in embryos exposed to 15 Gy. MO (IKH) was found to be higher than others at 13 Gy to 18 Gy but only observed at 13 Gy, 15 Gy and 16 Gy with the percentage of 22, 40 and 68%, respectively. Another abnormality, which is BMC (IA), was developed in embryos gamma irradiated with 15 Gy to 20 Gy and the percentage increased with increasing dose. However, BMC (IA) was observed as the highest abnormality at 20 Gy as compared to other abnormalities, with the percentage of 60%. BMC (IA) was found to be higher than IN group at 17 Gy to 20 Gy of gamma irradiation doses with the percentage of 38 to 60%, respectively.

Figure 5c shows the percentage of abnormalities in each treatment group at 72 h post-fertilization. MO (IN) was noted as the highest abnormality at 11 Gy as compared to other abnormalities with percentages of 78%. For AA (IN), it was found in embryos irradiated with 12 Gy to 20 Gy and has been observed only at 12 Gy, 13 Gy and 15 Gy to 18 Gy of irradiation dose with percentages of 78, 82, 35, 37, 20 and 18%, respectively. Based on the data 45% of O (IKH) was only observed at 12 Gy of irradiation dose while AA (IKH) were observed at 13 Gy, 15 Gy, 16 Gy and 20 Gy with the percentages of 40, 70, 83 and 33%, respectively. AA (IA) was the highest abnormality found at 20 Gy as compared to others with a percentage of 60%. Additionally, 17 and 25% of AA (IA) were also observed in zebrafish irradiated with 15 Gy and 16 Gy of gamma rays. AA (IA) was found in zebrafish irradiated with 17 Gy to 19 Gy as well. The percentage of this abnormality increased with increasing gamma irradiation doses. Abnormality in IKH and IA groups were found to be higher than IN group at 14 Gy to 20 Gy and 17 Gy to 20 Gy, respectively.

Figure 5d depicts the percentages of developed abnormalities in zebrafish embryos from the treatment groups for each irradiation dose. The results indicated that AA (IN) abnormality was observed in embryos irradiated with 11 Gy to 13 Gy, 18 Gy and 19 Gy of gamma rays with the percentage ranging from 18% to 82%. AA (IN) was also higher than other abnormalities at 11 Gy to 14 Gy. MCO (IKH) was found in embryos irradiated with 12 Gy to 17 Gy and significantly demonstrated at 12 Gy, 13 Gy and 16 Gy with the percentages of 45, 35 and 55%, respectively. No MCO (IKH) was recorded in embryos after irradiation with 18 Gy and 19 Gy gamma rays. MCO (IKH) was found to be lower than AA (IN) at irradiation doses of 12 Gy to 14 Gy. In addition, 33% of AA (IKH) was noted in embryos irradiated with 20 Gy of gamma rays, and it was also found in irradiated embryos at 13 to 19 Gy. AA (IKH) was found to be lower than AA (IN) at 13 Gy to 16 Gy of irradiation dose. MCO (IA), on the other hand, has been reported as the highest abnormality in irradiated embryos at 20 Gy of gamma radiation. This abnormality can be found in irradiated embryos given doses of 15 Gy to 19 Gy and the percentages increased with increasing irradiation doses. No MCO (IA) was identified at 11 Gy to 14 Gy and it was observed to be lower than AA (IN) at irradiation doses of 15 Gy and 16 Gy. The results also indicated that abnormalities in the embryos in IN group occurred at the dose of 11 Gy to 20 Gy. Abnormalities in IKH group were manifested at doses of 12 Gy to 20 Gy, whereas abnormalities in IA group were observed from those given gamma irradiation doses of 15 Gy to 20 Gy.

#### Abnormality Analysis in the Zebrafish Embryos

An investigation of the abnormalities that developed in embryos from the treatment groups was conducted. This enables differentiation amongst the abnormalities developed in embryos according to the treatments applied for each irradiation dose and different growth stage as well. For zebrafish in the IN treatment group, the percentages of abnormalities are depicted in Figure 6a. From the results, the embryos developed BM and M abnormalities at 24 h post-irradiation (48 hpf embryos). There were 32% and 25% of M developed in irradiated embryos at 14 Gy and 15 Gy while 28%, 35%, 30%, 40% and 25% of embryos developed BM after irradiated with 16 Gy to 20 Gy, respectively. Additional abnormality was discovered, which was pericardial oedema at 48 h post-irradiation (72 hpf embryos), when MO developed in the irradiated embryos at doses of14 Gy and 15 Gy. The percentage of BM in irradiated embryos at 16 Gy remained the same at 48 h post-irradiation. However, the percentages decreased for irradiated embryos at 17 Gy to 20 Gy with the percentages of 20%, 18%, 27% and 8%. At 72 h post-irradiation, AA developed at 12 Gy to 20 Gy with the percentages of 8% to 82%. Other abnormality developed was MO at 11 Gy (78%). After 96 h of irradiation, 78% of zebrafish embryos in IN group that developed MO earlier, experienced all of the abnormalities (AA). In addition, the percentages of AA in embryos irradiated with other doses (12 Gy to 20 Gy) remained the same at this stage.

Abnormalities in the IKH group were different as compared to other treatment groups. Only microphthalmia (M) abnormality was detected at 24 h post-irradiation in irradiated embryos at 17 to 20 Gy. It was only noted as the highest abnormality at 20 Gy with the percentage of 55%. Subsequently, the embryos from IKH group developed MO at 48 h post-irradiation with the percentage increasing from 22% to 68% at 13 Gy to 16 Gy, respectively. MO also developed in embryos irradiated at doses of 17 Gy to 20 Gy with percentages of 42%, 47%, 37% and 33%, respectively. However, among the irradiation doses, MO was only noted at 13 Gy. There were two types of abnormalities developed in the embryos at 72 h post-irradiation viz. pericardial oedema (O) and all of the abnormalities (AA). The O abnormality was observed only for irradiation dose of 12 Gy, while AA was found in embryos irradiated at 13 Gy to 20 Gy. AA (72) was noted in embryos irradiated at 14 Gy to 16 Gy with the percentage of 52% to 83%, respectively. Furthermore, at this 72 h post-irradiation, the percentage of AA abnormality was observed to increase from 13 Gy to 16 Gy, with percentage of 40% to 83%, respectively. The percentage then reduced at irradiation dose of 17 Gy with percentage of 42%. At 96 h post-irradiation, MCO and AA abnormalities were found in the irradiated embryos. MCO (96) was found in zebrafish embryos irradiated at 12 Gy to 17 Gy and AA (96) was identified at 13 Gy to 20 Gy. Additionally, AA was observed at 14 Gy to 16 Gy with percentages of 15%, 20% and 28%. Percentage of AA (96) increased from 13 Gy to 18 Gy (5% to 47%) in the study, and it was found to be lower about 22% to 30% than MCO (96) at 13 Gy to 16 Gy. Figure 6b depicts the percentage of abnormalities identified in the irradiated zebrafish embryos from the treatment groups.

Figure 6c shows the abnormalities developed in irradiated zebrafish embryos at 24 to 96 h-post irradiation. From the results, no abnormality was found in irradiated zebrafish at 11 to 14 Gy. At 24 h post-irradiation, BMC recorded in embryos at 17 Gy to 20 Gy and noted as the lowest abnormality at 15, 16, 17, 18 and 20 Gy with the percentages of 0% to 45%. Percentage of BMC at 48 h post-irradiation (BMC (48)), AA at 72 h post-irradiation (AA (72)) and MCO at 96 h post-irradiation (MCO (96)) were found to be similar at 15 to 20 Gy besides increased with increasing the irradiation dose. The percentages of abnormalities in IA group at 48 to 96 h post-irradiation were 17 to 60% from 15 to 20 Gy of irradiation dose.

### 2.3. DNA Damage Evaluation

Figure 7 depicts the intensity of γ-H2AX expression in the non-irradiated (control) and irradiated zebrafish embryos. The figure also shows the γ-H2AX expression in each group at 11 Gy, 15 Gy and 20 Gy to differentiate clearly between the effects of every irradiation dose. Figure 8 shows the percentage of γ-H2AX intensity in the irradiated and non-irradiated zebrafish embryos. Based on the results, γ-H2AX intensity in all of the irradiated embryos were recorded to be higher than the control at 11 Gy to 20 Gy with percentage of 34% to 65% (Figure 8a). Furthermore, untreated embryos were noted to be higher than the treatment groups at 11 Gy to 19 Gy with the percentage of 20%to 65% (Figure 8b). The lowest intensity for each irradiation dose was noted in 34% to 37% and 39.6% to 40.1% of irradiated zebrafish embryos pre-treated with amifostine at 11 Gy to 14 Gy and 17 Gy to 20 Gy, respectively (Figure 8b). No significant difference was recorded between irradiated embryos pre-treated with KH and amifostine at 15 Gy and 16 Gy with percentage of 37% to 43%, respectively.

### 2.4. Apoptosis Evaluation

Figure 9 depicts the intensity of caspase-3 expression in the non-irradiated (control) and irradiated zebrafish embryos. The figure also shows the caspase-3 expression in each group at gamma irradiation doses of 11 Gy, 15 Gy and 20 Gy to differentiate clearly between the effects of every irradiation dose. Figure 10 shows the percentage of caspase-3 intensity in the irradiated and non-irradiated zebrafish embryos. Based on the results, caspase-3 intensity in the untreated irradiated embryos and irradiated embryos pre-treated with KH were noted to be higher than control at 11 Gy to 20 Gy with percentage of 51% to 62% and 37% to 59%, respectively (Figure 10a). Furthermore, zebrafish embryos pretreated with amifostine were noted to be higher than the control at 13 Gy to 20 Gy with the percentage of 28% to 40% (Figure 10a). The lowest intensity for each irradiation dose was noted in 25% to 40% of irradiated zebrafish embryos pre-treated with amifostine at 11 Gy to 20 Gy (Figure 10b). Caspase-3 intensity in irradiated embryos pre-treated with KH were also noted to be lower than untreated irradiated embryos at 11 Gy to 19 Gy with the percentage of 37%to 59% (Figure 10b).

## 3. Discussion

### 3.1. Lethality Analysis in Zebrafish Embryos

From the results, amifostine treatment was observed to reduce more lethality in the irradiated zebrafish embryos as compared to the untreated and KH treatment groups. It is evident that amifostine is able to reduce lethality from ionizing radiation, particularly gamma rays in the present study. The result was in accordance with Koukourakis et al. [23], Daroczi et al. [24] and McAleer et al. [25], who reported that amifostine was demonstrated to improve survival of irradiated zebrafish embryos. Therefore, amifostine, a known radioprotectant, has been selected in this study as positive control as well as a reference agent to investigate the potential of KH to reduce harmful effects of ionizing radiation. Koukourakis et al. [23] also used amifostine as a positive control to evaluate radioprotection properties of fullerene nanoparticle DF-1 in zebrafish model.

KH treatment on the zebrafish embryos was shown to reduce the lethality effects at 24 h post-irradiation up until the highest irradiation dose (20 Gy). However, at 48 h post-irradiation, KH was ineffective at reducing the lethality effects when exposed to the higher range of irradiation doses (17 to 20 Gy). That was a sign of severe effects by ionizing radiation at higher doses which could not be reduced by radioprotecting agents. The effectiveness of any radioprotectors is limited to a threshold that cannot exceed the radio-sensitizing action of oxygen dissolved in tissues, which increases radiation damage of the cells by 2- to 3-fold [26]. However, the accumulated number of coagulated zebrafish embryos for each irradiation dose in the IKH group was lower than untreated group and this reflects on the efficacy of KH to reduce overall lethality effects. The results were in accord with the study by Jagetia [27], which reported that there were several natural products able to improve survival of subject in in vivo model.

### 3.2. Abnormality Analysis in Zebrafish Embryos

There were combinations of abnormalities developed in each different growth stage among all the treatment groups. This is similar to the findings of McAleer et al. [25], who reported that irradiation of zebrafish embryos produced multiple morphologic defects considered as common, nonspecific manifestations of embryonic mutagenesis. In addition, Lu et al. [28] stated that such radiation-induced malformations show similarity to those noted in mammals. Exposure to ionizing radiation resulted in cataract formation, retinal degeneration/atrophy, blindness, and microcephaly. The eyes were markedly affected (microphthalmia) from the many abnormalities seen in the zebrafish embryos at 24 and 48 h post-irradiation. This indicated that the organization of retinal cellular layers was noticeably perturbed. At 72 h post-irradiation, most of the surviving embryos from the treatment groups developed all of the abnormalities (body curvature, microphthalmia, microcephaly and pericardial oedema). It means more severe harmful effects occurred in the embryos at this juncture. This result was consistent with the study by Geiger et al. [29], where morphologic abnormalities of the embryos may change over time and depend on the radiation doses. More severe abnormalities developed in the irradiated zebrafish embryos led to lethality. This can be seen from the results where the abnormalities reduced with increasing the irradiation doses. More embryos died as irradiation dose reached 20 Gy. The proportion of lethality was higher than abnormalities for untreated zebrafish embryos.

The mechanisms of radioprotection may depend on the cellular or tissue-type merits confirmation, especially in other animal models [29]. The results show that amifostine reduced the abnormalities in the irradiated zebrafish embryos (up to 17 Gy) at 24 h post-irradiation. However, the protective effects decreased to 16 Gy at 48 h, 72 h and 96 h post-irradiation. To further describe this radioprotective events of amifostine, Geiger et al. [29] once again suggested that it may be possible that the ability of different cell types to tolerate radiation may be affected by radiomodifiers (able to modify radiation effects) due to differences in the repair capacity, in the intracellular pathways, that are affected, or in the ability to absorb the radiomodifying agent. At organ level, there may be variations in perfusion, pH status, or hypoxia, any or all of which may influence the efficacy of the amifostine [30]. Radiation doses of more than 17 Gy were considered ineffective for amifostine as radioprotective agent. This is because of a threshold dose of radiation, beyond which amifostine is largely ineffective [29]. Referring to the results, most of the zebrafish embryos from all of the treatment groups developed AA (all of the abnormalities) at 72 h post-irradiation. However, for amifostine group, the body curvature abnormality was ameliorated by the radioprotectant at 96 h post-irradiation. Only MCO abnormality was found instead of AA at 72 h post-irradiation. Amifostine may induce organ-specific levels of radioprotection, causing the agent to be more effective in preventing the radiation-induced death of specific organs [30].

For KH, the radioprotective effects varied according to the developmental stages of the zebrafish embryos. KH was observed to radioprotect the embryos receiving up to 17 Gy of gamma irradiation, at 24 h post-irradiation and the protective effects were reduced at 48 h, 72 h and 96 h post-irradiation. KH has been established as a potential therapeutic antioxidant agent for various biodiverse diseases and toxicants [31,32]. An investigation was done by Biluca et al. [33] on several honey samples from nine different species of stingless bee honey. They found that the honey samples contained 26 phenolic compounds (antioxidants), which played important roles in protection against oxidative stress by free radicals. Phenolic acids and flavonoid are the components of the phenolic compounds and responsible to inhibit free radicals, inhibiting enzymatic systems that produce free radical forms, increasing the concentration of biologically important endogenous antioxidants, and inducing the expression of a variety of genes responsible for the synthesis of enzymes that inhibit oxidative stress [34]. The efficiency of phenolic compounds in protection against oxidative stress depended on their reactivity in relation to toxic oxygen species and the reactivity of phenoxy radicals relative to critical biomolecules [34]. Massive generation of free radicals by ionizing radiation can result in the decline of any intracellular protection mechanism, affecting any place in the cell, and with an intensity that depends on the dose rate of absorbed radiation as well as the linear transfer of energy of the radiation administered [35]. In this study, the dose of 13 Gy and above might induce massive generation of free radicals that exceed antioxidants protective mechanism by KH and resulted in morphologic abnormalities of the zebrafish embryos. However, similar to amifostine, KH was seen to induce organ-specific levels of radioprotectant by ameliorating the body curvature abnormality at 96 h post-irradiation, which was the endpoint of the phenotypic study. It can be seen by the reduction in the percentages of AA from 72 h post-irradiation to 96 h post-irradiation in the IKH group. Thus, having a potential to reduce body curvature suggested it as an agent that can reduce apoptosis of neural cells in developing spinal cord.

### 3.3. DNA Damage Analysis

Gamma irradiation was confirmed to cause DNA damages in zebrafish embryos in comparison to the negative control. It can be seen by the expression of γ-H2AX protein in the zebrafish embryos from the treatment groups. DNA damages are always associated with double stranded breaks and were often followed by the phosphorylation of the histone, H2AX [36]. H2AX is a variant of the H2A protein family which is phosphorylated by kinases such as ataxia telangiectasia mutated (ATM). Thus, newly phosphorylated protein which is γ-H2AX produced as a novel biomarker for DNA damage. Double stranded breaks caused by ionizing radiation can be identified and quantified in situ by detecting the γ-H2AX foci formed at DNA break sites utilizing the immunostaining assay in this study [34]. These foci represent the double stranded breaks in a 1:1 manner as well as γ-H2AX intensity and are the most sensitive for the current assay.

Furthermore, this immunostaining assay was performed to confirm the effectiveness of radioprotectant agents in this study particularly KH. Amifostine and KH both reduced DNA damages in this study by reducing the percentages of γ-H2AX expression in every tested radiation doses. The percentages of DNA damages in zebrafish embryos treated with amifostine were much lower than KH. Hofer et al. [37] reported that amifostine treatment protect normal human dermal fibroblasts (NHDF) fibroblasts from radiation damage and also support double stranded repair in the cells by using comet assay and γH2AX/53BP1 foci immunofluorescence assay. In vivo study in murine by González et al. [38] found that amifostine significantly reduced the DNA damage in cells from the animals pre-treated with amifostine. Amifostine’s protection properties will be activated after dephosphorylation to WR-1065, its active and free radical-scavenging sulfhydryl metabolite [37]. WR-1065 is a thiol form of amifostine that participates in direct chemical repair via the donation of hydrogen atoms and the depletion of oxygen to a single state following thiol oxidation, and the induction of cellular hypoxia [39]. Such event leads to a decrease in the number of DNA double-strand breaks [39].

KH also was seen to effectively exhibit protective effects by reducing DNA damages in the embryos. Antioxidant contents in KH saved cells from the harmful effects of reactive oxygen species induced by gamma radiation. The free radicals trigger the deterioration of membranes, lipids, amino acids and DNA [40]. This finding was supported by Misra et al. [41] that reported stingless bee honey (SBH) from *Trigona* sp. enhanced anti-oxidant defenses, upregulated Nrf2 (protein that regulates the expression of antioxidant proteins), and reduced lipid peroxidation and oxidative DNA damage in in vivo study using Sprague Dawley rats. In addition, an in vitro study conducted by Abdul Karim [42] found that KH reduced the oxidative stress (precursor to DNA damage) in lymphoblastoid cell line (LCL) through its free radical scavenging activity (phenolic and flavonoid compounds). However, the protection to DNA damages by KH was only limited to irradiation doses below 20 Gy (11 Gy–19 Gy). This could be due to hormesis effects, where by high doses of ionizing radiation are harmful and cannot be protected by any radioprotective mechanism.

### 3.4. Apoptosis Analysis

In the immunostaining study, capase-3 was used to evaluate the apoptosis effects in the irradiated embryos from the treatment groups. Caspase-3 is known as an executioner caspase in apoptosis because of its function in coordinating the damage of cellular structures such as DNA fragmentation or degradation of cytoskeletal proteins. The activity of caspase-3 is tightly controlled and it is produced as zymogen in an inactive pro-form. Caspase-3 antibodies serve as excellent biomarkers to monitor induction of apoptosis by detecting the levels of pro caspase-3 and its active form. Cleavage and activation of pro caspase-3 is catalyzed by caspase-8, caspase-9, and granzyme B to generate the active heterodimer of caspase-3 subunits [43]. The results also show that increase in radiation doses caused the increased in caspase-3 intensity. Dos Santos et al. (2017) [44] suggested that the response of active caspase-3 expression to radiation induced damage is dose-dependent, but varies significantly among people and so determined by individual characteristics. As a secondary response to DNA damage [11], apoptosis also is promoted by ROS in the cellular death signalling pathway. ROS is believed to play an important role in the promotion of apoptosis by affecting mitochondrial permeability, release of cytochrome c, activation of p53 and the action of caspases [45].

The irradiated embryos from untreated and pre-treated with KH exhibited higher caspase-3 intensity as compared to the negative control group from 11 Gy to 20 Gy of gamma irradiation. In contrast to other treatment groups, embryos pre-treated with amifostine exhibited higher caspase-3 intensity as compared to the negative control started at 13 to 12 Gy. Amifostine is suggested to up-regulate the production of inhibitor of apoptosis (IAP) protein in the embryos irradiated at 11 Gy and 12 Gy from the treatment group. Interaction of IAP proteins will inhibit the enzymatic activity of both initiator and effector caspases [46]. Several important mammalian IAPs including XIAP, c-IAP1, c-IAP2 and ML-IAP have been identified in previous studies and all of them exhibited anti-apoptotic activity in cell culture [47]. Caspase-3 is one of the effector caspases which plays a role in cleaving numerous target proteins, broadly distributed throughout the cell, resulting in the morphological changes that are characteristic of apoptosis [48]. Santoro et al. [49] reported that c-IAP1 deficiency in zebrafish embryos leads to a massive apoptosis in endothelial cells, through a regulation of TNFR signalling.

In this study, embryos from both amifostine and KH groups showed reduction in caspase-3 expression in comparison to the untreated group. This could explain their radioprotective ability to reduce apoptosis induced by ionizing radiation exposure. The capability of amifostine to reduce apoptosis was in accordance with Geiger et al. [29] who found that the known radioprotectant reduced apoptotic effects in the eyes and brain of zebrafish embryos. A study by Mercantepe et al. [50] found that the expression of caspase-3 decreased in spermatogonial cells treated with amifostine. Even higher than amifostine, the caspase-3 intensity of embryos in pre-treated with KH was also significantly decreased in the study. It was the evidence that showed the ability of KH to reduce apoptosis effects induced by ionizing radiation. In vivo study by Aziz et al. [51] found that rats receiving stingless bee honey supplementation reduced apoptotic events induced by oxidative stress in the pancreas. Additionally, KH treatment in rats was found to increase SOD (antioxidant enzyme) that may have increased superoxide conversion into hydrogen peroxide which reduces superoxide anion levels to prevent osteoblast and osteocytes apoptosis while reducing osteoclast production [52]. The ability of amifostine and KH to reduce apoptosis is related to their antioxidant capacities in scavenging free radicals. However, higher doses of ionizing radiation, for example 20 Gy and above may cause depletion of an antioxidant active form, leading to increasing risk of apoptosis. Radiation at doses used in therapy depletes cellular alpha tocopherol in normal cells, thereby increasing their risk of damage [53].

## 4. Materials and Methods

### 4.1. Zebrafish Care and Embryo Collection

Adult zebrafish (4 to 5 months old) were purchased from Danio Assay Laboratories Sdn. Bhd. (Selangor, Malaysia) and raised according to the Institutional Animal Care and Use Committee (IACUC) of Universiti Putra Malaysia (Ethic Approval Reference Number: UPM/IACUC/AUP-R065/2016). There were 16 males and eight females of AB wild-type strain zebrafish kept and maintained in an aquarium tank (still tank) in the laboratory. The regular care and maintenance of the zebrafish were adapted from Organisation for Economic Cooperation and Development [54]. Water temperature in the tank was maintained at 26 °C ± 1 °C [54] and the photoperiod was regulated at 14:10 h (light:darkness) [55]. The fish were fed three times daily with a mixture of Hikari Micropelets and Hikari Microwafers (city, Japan). The adult zebrafish with a ratio of 2 (male):1 (female) were placed in the still tank [54]. Sufficient air was supplied through an external pump without any disturbance in the flow of water. For production of eggs, a specialized egg/embryo collection box was placed in the aquarium tank a day before use. Marbles and artificial plants were provided to stimulate spawning of the fish [54]. The eggs were collected from the tank 3 h after the onset of light. The viable normal dividing spherical eggs were washed and placed in a petri dish containing 1× E3 medium at 26 °C ± 1 °C. The eggs (zebrafish embryos) were selected by using a stereomicroscope (Labomed, LA, USA).

### 4.2. Embryo Dechorionation

The embryos at 24 hpf were placed in the 1× E3 medium with 1 mg/mL pronase (Roche, Mannheim, Germany) for five minutes at room temperature, then gently agitated with a plastic pipette until the embryos were freed and kept in Petri dishes containing E3 medium at 26 ± 1 °C [56]. Dechorionation for zebrafish embryos at 24 hpf above is important to improve any treatment or test in zebrafish [56]. The dechorionated embryos were washed and placed into 60 × 15 mm size petri dish containing 1× medium.

### 4.3. Amifostine or KH Pretreatment

A modification of McAleer et al. [25] protocols was adapted for this experiment. In this study, the embryos were divided into four groups. There were two untreated groups including untreated and non-irradiated embryos as well as untreated embryos underwent irradiation (IN). For treatment groups, there were two categories, namely embryos pre-treated with KH (IKH) and embryos pretreated with amifostine (IA). Amifostine and non-irradiated embryos were used as positive and negative control respectively, in this radioprotective study. Amifostine was purchased from Merck (Darmstadt, Germany) and Kelulut honey (KH) was provided by Marbawi Food Trading and Processing (Kuala Kangsar, Perak, Malaysia). Treatment doses used in this study were 4 mM [25] and 8 mg/mL for amifostine and KH, respectively. The KH dose was selected according to LC_10_ (concentration at which 10% of lethality was observed) value from our previous works on toxicity of KH [57]. The dechorionated zebrafish embryos at 24 hpf were treated with amifostine or KH for 30 min before gamma radiation exposure.

### 4.4. Gamma Irradiation

The zebrafish embryos at 24 hpf were selected by using a stereomicroscope (Labomed). The dechorionated embryos were washed and placed into 60 × 15 mm size petri dish containing 1× E3 medium. The gamma irradiation method was according Mena et al. [58]. The embryos in IN, IKH and IA groups were irradiated at the doses of 11 Gray (Gy) to 20 Gy. These doses were selected based on our previous radiosensitivity study on the zebrafish embryos. They were irradiated in a single fraction of gamma rays coming from a Gamma Cell BIOBEAM GM 8000 ^137^Cs (caesium-137) source (Gamma-Service Medical GmbH, Leipzig, Germany) at a dose rate of about 5 Gy/min. For the irradiation procedures, the Petri dish containing zebrafish embryos was fixed at centre position on six layers of polystyrene in a stainless steel BB75-4 beaker, as shown in Figure 11. The stainless steel beaker was then inserted into a chamber in the Gamma Cell BIOBEAM GM 8000 for irradiation. All of the untreated and treated embryos were incubated at 26 ± 1 °C after the irradiation procedure. The negative control embryos were also been incubated at the same temperature and incubation time. Lethality and abnormalities in the embryos were recorded at 24, 48, 72 and 96 h post-irradiation.

### 4.5. Survival Analysis

The 24-well plates containing zebrafish embryos from all of the embryos groups were incubated at 26 °C ± 1 °C. Assessment of toxicological endpoints (lethality) was performed using stereomicroscope (Labomed) at 24 h, 48 h, 72 h and 96 h post-irradiation. Lethality in the study was recorded based on the number of zebrafish embryos that coagulated and lacked somite formation with non-detachment of the tail and without cardiac pulse [52]. Graphs that visualized the percentage of mortality versus radiation absorption dose (gray, Gy) were plotted.

### 4.6. Morphological Analysis

The 24-well plates contained zebrafish embryos from all of the treatment groups were incubated at 26 ± 1 °C. Abnormalities that include body curvature, changes of eye (microphthalmia), head (microcephaly), pericardial oedema, and haemorrhagic occurrence of the irradiated zebrafish embryos were recorded every 24 h. Observations were made using a Labomed stereomicroscope. Graphs that visualized the percentage of abnormalities versus radiation absorption dose (gray, Gy) were plotted.

### 4.7. Immunohistochemistry Analysis

Similar to lethality and abnormality effects study, all of the embryos were divided into four groups for this immunohistochemistry (IHC) analysis [59]. In this analysis, the protein expressions of DNA damages and apoptosis were investigated. After irradiation procedure, the embryos were fixed in 4% paraformaldehyde (PFA) in phosphate buffered saline (PBS) overnight at 4 °C and dehydrated in 100% methanol at −20 °C. The embryos then were rehydrated through a series wash of methanol in phosphate buffer saline tween 20 (PBST) (100%, 75%, 50% and 25% in sequence). The embryos were then washed for 3 times with PBST. To maintain a stable pH, Tris buffer was added into the tubes containing embryos. The embryos were also equilibrated in the tris buffer for 15 min. After equilibration, the embryos were washed with PBST and rinsed with distilled water for twice. For this IHC analysis, acetone treatment is used after equilibration to permeabilize the embryos membrane. Before antibody treatment, the embryos were blocked with freshly prepared 2% Phosphate Buffer Triton X (PBT) to prevent non-specific binding of the antibodies as well as reduce the background formation that interrupt the imaging process. The incubation period in the blocking solution was four h. Then, purified polyclonal antibodies γ-H2AX (1:200; GeneTex, CA, USA) for DNA damage or caspase-3 (1:200; BD Biosciences, CA, USA) for apoptosis were added and incubated overnight at 4 °C. On the next day, the embryos were washed 4 times with PBT for 30 min each and incubated with Alexa-488–conjugated anti-rabbit IgM secondary antibody (1:200; Invitrogen, CA, USA) incubated for 3 h at room temperature. The embryos were then washed five times with PBST and fixed with 4% PFA for 20 min at room temperature. Following the fixation was washing again with PBST and finally, before fluorescence microscope observation, the embryos were immersed with series of glycerol/PBS solution (25% to 100%). The expression of proteins γ-H2AX and caspase-3 were observed using fluorescence microscopy (Nikon, Tokyo, Japan) with a 490-nm filter.

### 4.8. Data Analysis

Protein expression intensities (γ-H2AX and caspase-3) were measured in percentage by ImageJ analysis software (University of Wisconsin, Madison, WI, USA). All the data were analyzed using one-way ANOVA followed by Dunnett’s and Duncan post hoc test using Social Science (SPSS) version 21 (IBM, Chi, USA) and presented as mean ± SEM, where *p* < 0.05 was considered as significant.

## 5. Conclusions

In conclusion, KH has been demonstrated to increase the survival rate of zebrafish embryos as it possesses protective properties from ionizing radiation exposure. KH was also proven to protect zebrafish embryos from organ-specific abnormality which is body curvature. Furthermore, KH was shown to exhibit cellular protective mechanism by reducing DNA damage and apoptosis through the selected proteins, γ-H2AX and caspase-3 evaluations. These results have broad implications for a fast screening of novel radiation protectors and sensitizers for potential human therapeutic use. 

## Figures and Tables

**Figure 1 molecules-26-01557-f001:**
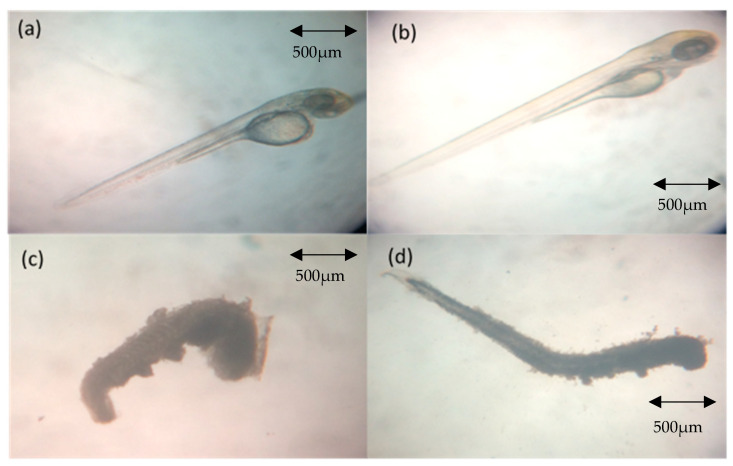
Normal and coagulated zebrafish embryos. (**a**) At 48 h post fertilization (hpf) (24 h post-irradiation) from negative control group (normal development); (**b**) at 72 hpf (48 h post-irradiation) from negative control group (normal development); (**c**) at 48 hpf (24 h post-irradiation) from the treatment groups (coagulated); (**d**) at 72 hpf (48 h post-irradiation) from the treatment groups.

**Figure 2 molecules-26-01557-f002:**
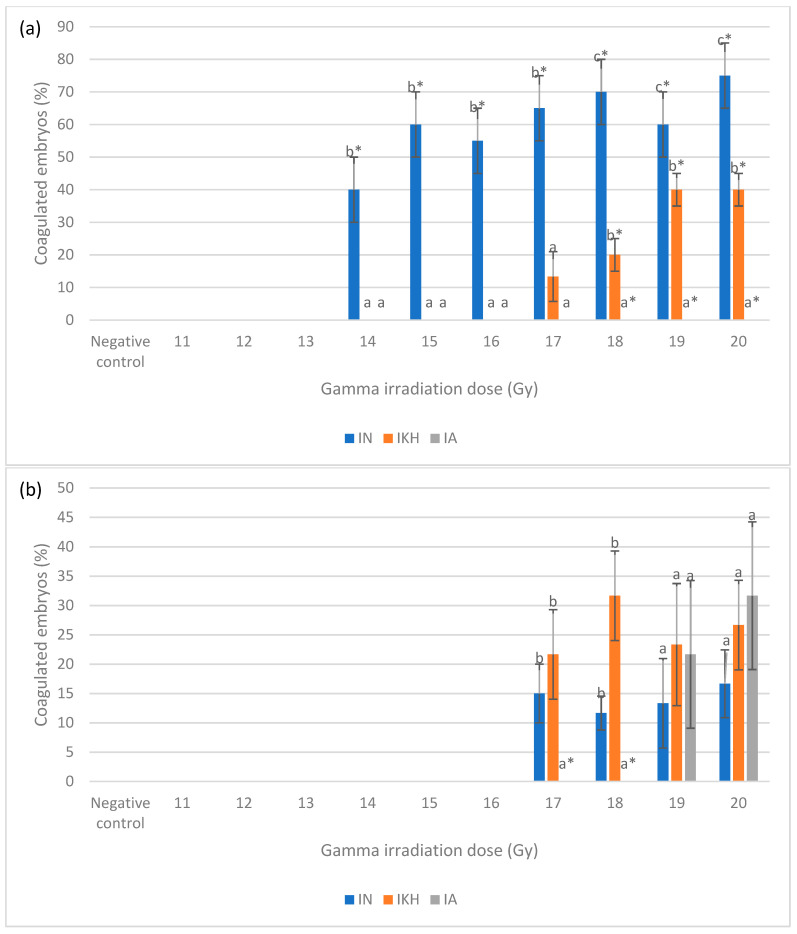
(**a**) Coagulated zebrafish embryos in IN, IKH and IA groups at 24 h post-irradiation. (**b**) Coagulated zebrafish embryos in IN, IKH and IA groups at 48 h post-irradiation. a*, b* and c* indicate significant difference (*p* < 0.05) compared to other treatment groups. a and b indicate no significant difference (*p* > 0.05) compared to other treatment groups.

**Figure 3 molecules-26-01557-f003:**
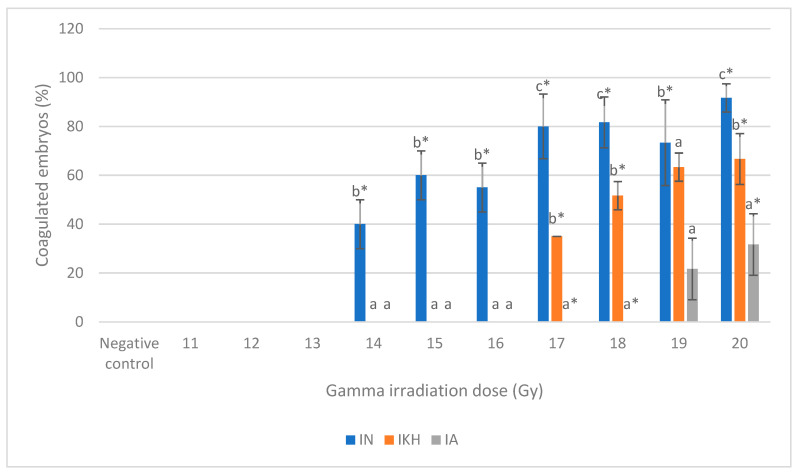
Total coagulated zebrafish embryos in IN, IKH and IA group. a*, b* and c* indicate significant difference (*p* < 0.05) compared to other treatment groups. a indicates no significant difference (*p* > 0.05) compared to other treatment groups.

**Figure 4 molecules-26-01557-f004:**
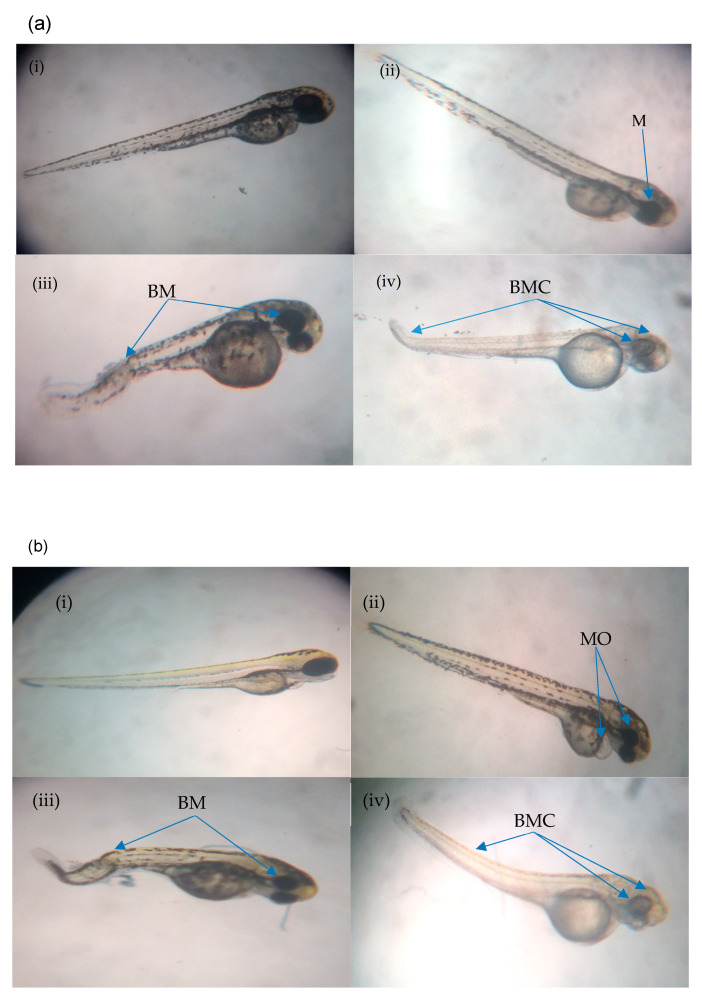
(**a**) Normal and abnormal zebrafish embryos at 24 h post-irradiation. (**a** (i)) negative control (normal development); (**a** (ii)) abnormal morphology in IN and IKH groups (microphthalmia); (**a** (iii)) abnormal morphology in IN group (body curvature with microphthalmia); (**a** (iv)) abnormal morphology in IA group (body curvature with microphthalmia and microcephaly). (**b**) Normal and abnormal zebrafish embryos at 48 h post-irradiation. (**b** (i)) Negative control (normal development); (**b** (ii)) abnormal morphology in IN and IKH groups (microphthalmia with pericardial oedema); (**b** (iii)) abnormal morphology in IN group (body curvature with microphthalmia); (**b** (iv)) abnormal morphology in IA group (body curvature with microphthalmia and microcephaly). (**c**) Normal and abnormal zebrafish embryos at 72 h post-irradiation; (**c** (i)) negative control (normal development); (**c** (ii)) abnormal morphology in IN group (microphthalmia with pericardial oedema); (**c** (iii)) abnormal morphology in IKH group (pericardial oedema); (**c** (iv)) abnormal morphology in IN, IKH and IA group (body curvature with microphthalmia, microcephaly and pericardial oedema). (**d**) Normal and abnormal zebrafish embryos at 96 h post-irradiation; (**d** (i)) Negative control (normal development); (**d** (ii)) abnormal morphology in IN and IKH group (body curvature with microphthalmia, microcephaly and pericardial oedema); (**d** (iii)) abnormal morphology in IKH and IA group (microphthalmia with microcephaly and pericardial oedema).

**Figure 5 molecules-26-01557-f005:**
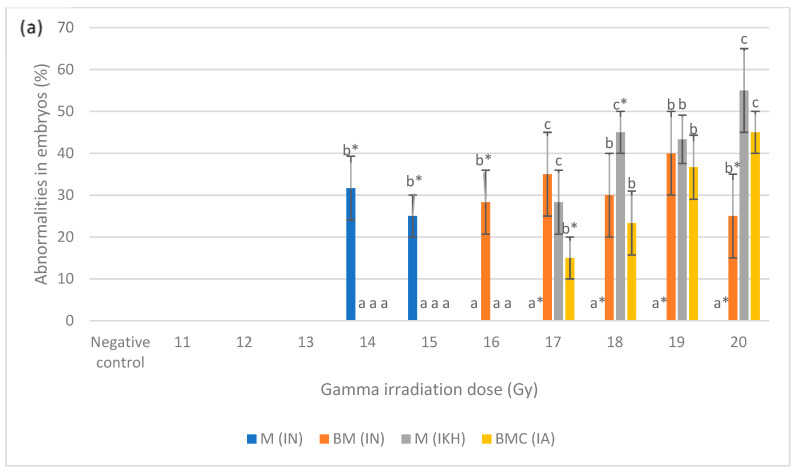
(**a**) Abnormalities produced for each dose in the irradiated zebrafish embryos at 24 h post-irradiation; (**b**) abnormalities produced for each dose in the irradiated zebrafish embryos at 48 h post-irradiation; (**c**) abnormalities produced for each dose in the irradiated zebrafish embryos at 72 h post-irradiation; (**d**) abnormalities produced for each dose in the irradiated zebrafish embryos at 96 h post-irradiation. a*, b*, c* and d* indicate significant difference (*p* < 0.05) compared to other abnormalities. a, b, c, ab and bc indicate no significant difference (p > 0.05) compared to other abnormalities. a, b, c, ab and bc indicate no significant difference (*p* > 0.05) compared to other abnormalities.

**Figure 6 molecules-26-01557-f006:**
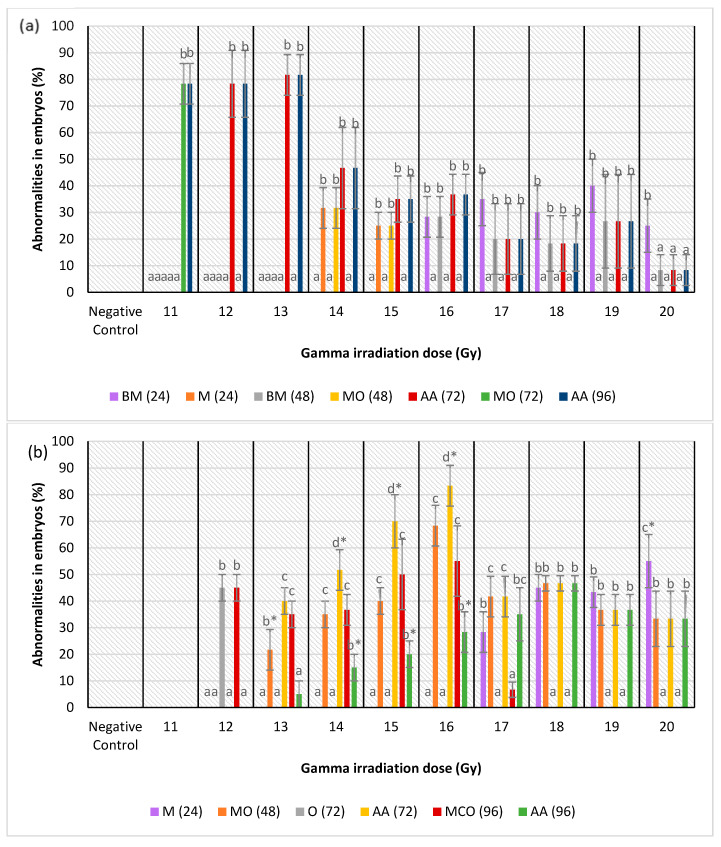
(**a**) Abnormalities produced for each dose in the irradiated zebrafish embryos in IN group; (**b**) abnormalities produced for each dose in the irradiated zebrafish embryos in IKH group; (**c**) abnormalities produced for each dose in the irradiated zebrafish embryos in IKH group. a*, b*, c* and d* indicate significant difference (*p* < 0.05) compared to other abnormalities. a, b, c and bc indicate no significant difference (*p* > 0.05) compared to other abnormalities.

**Figure 7 molecules-26-01557-f007:**
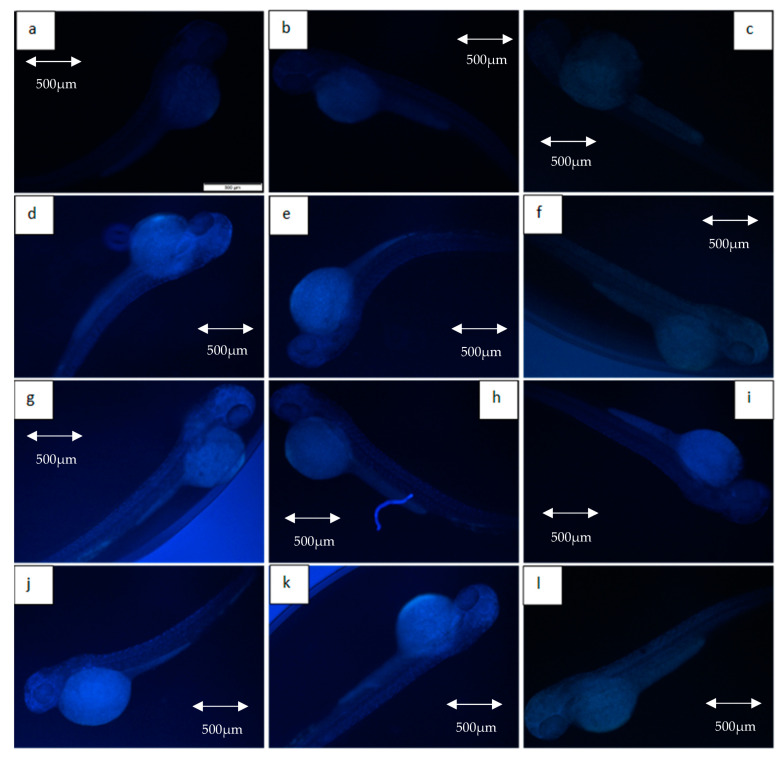
Expression of γ-H2AX in non-irradiated and irradiated zebrafish embryos. (**a**) Control without any treatment (non-irradiated); (**b**) control pre-treated with KH (non-irradiated); (**c**) control pre-treated with amifostine (non-irradiated); (**d**) untreated embryos irradiated at 11 Gy; (**e**) embryos pretreated with KH and irradiated at 11 Gy; (**f**) embryos pretreated with amifostine and irradiated at 11 Gy; (**g**) untreated embryos irradiated at 15 Gy; (**h**) embryos pretreated with KH and irradiated at 15 Gy; (**i**) embryos pretreated with amifostine and irradiated at 15 Gy; (**j**) untreated embryos irradiated at 20 Gy; (**k**) embryos pretreated with KH and irradiated at 20 Gy; (**l**) embryos pretreated with amifostine and irradiated at 20 Gy.

**Figure 8 molecules-26-01557-f008:**
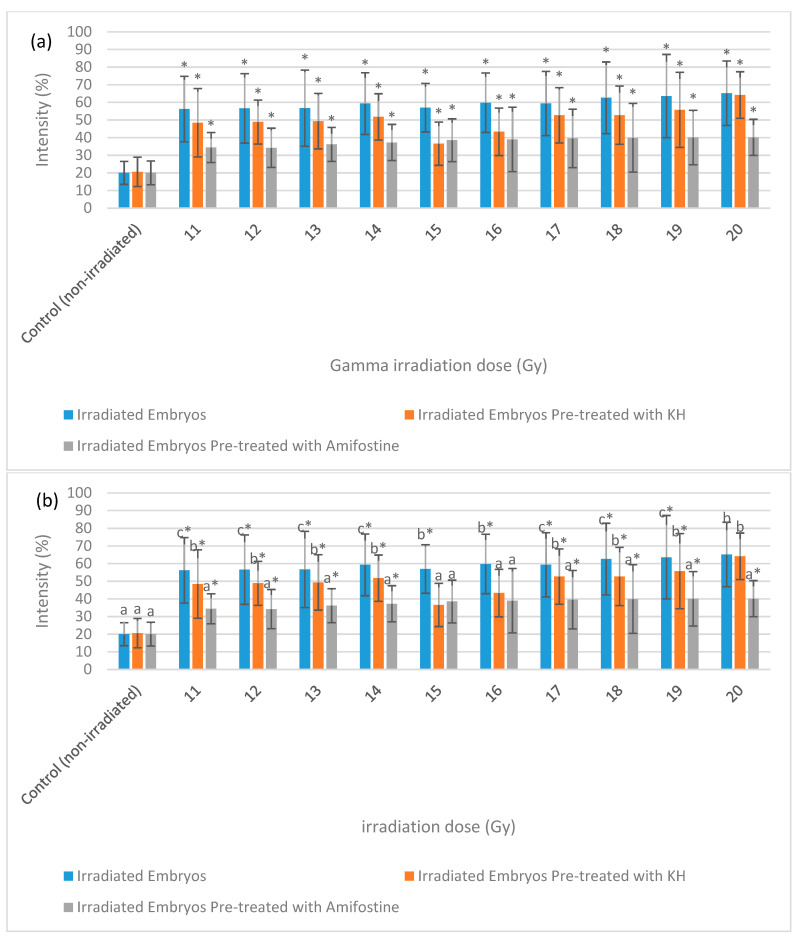
Intensity (%) of γ-H2AX in irradiated and non-irradiated zebrafish embryos. (**a**) γ-H2AX intensity in irradiated and non-irradiated embryos. (*) indicates significant difference (*p* < 0.05) as compared to control; (**b**) γ-H2AX intensity in untreated and embryos pre-treated with KH and amifostine. a*, b* and c* indicate significant difference (*p* < 0.05) compared to other treatment groups.

**Figure 9 molecules-26-01557-f009:**
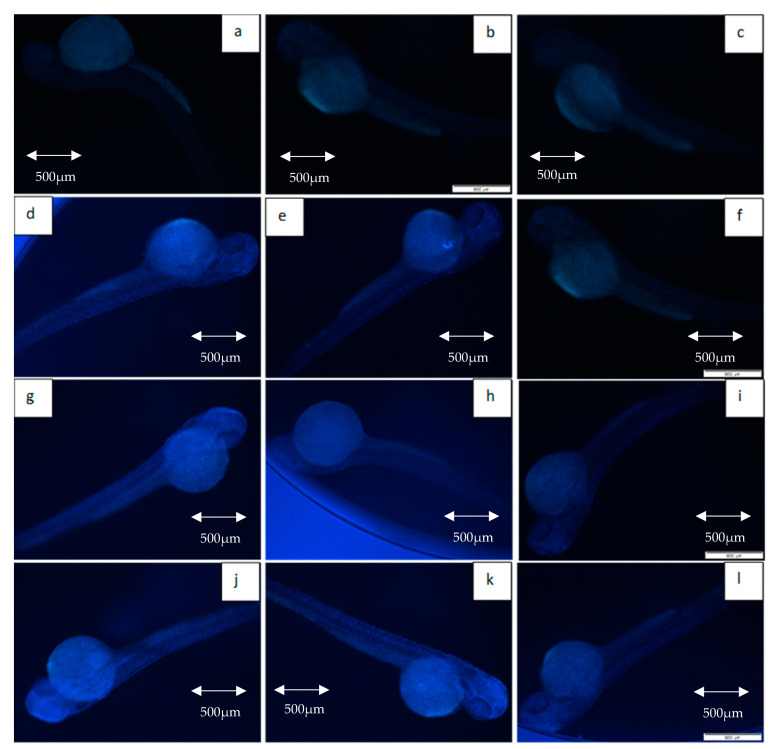
Expression of caspase-3 in non-irradiated and irradiated zebrafish embryos. (**a**) Control without any treatment (non-irradiated); (**b**) control pre-treated with KH (non-irradiated); (**c**) control pre-treated with amifostine (non-irradiated); (**d**) untreated embryos irradiated at 11 Gy; (**e**) embryos pretreated with KH and irradiated at 11 Gy; (**f**) embryos pretreated with amifostine and irradiated at 11 Gy; (**g**) untreated embryos irradiated at 15 Gy; (**h**) embryos pretreated with KH and irradiated at 15 Gy; (**i**) embryos pretreated with amifostine and irradiated at 15 Gy; (**j**) untreated embryos irradiated at 20 Gy; (**k**) embryos pretreated with KH and irradiated at 20 Gy; (**l**) embryos pretreated with amifostine and irradiated at 20 Gy.

**Figure 10 molecules-26-01557-f010:**
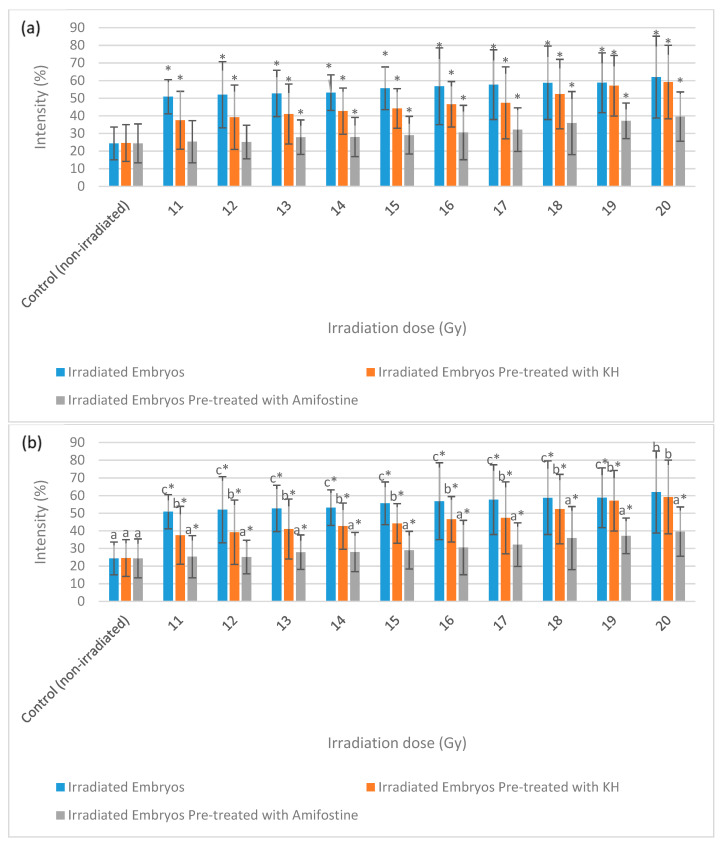
Intensity (%) of caspase-3 in irradiated and non-irradiated zebrafish embryos. (**a**) Caspase-3 intensity in irradiated and non-irradiated embryos. (*) indicates significant difference (*p* < 0.05) as compared to control; (**b**) caspase-3 intensity in untreated and embryos pre-treated with KH and amifostine. a*, b* and c* indicate significant difference (*p* < 0.05) compared to other treatment group.

**Figure 11 molecules-26-01557-f011:**
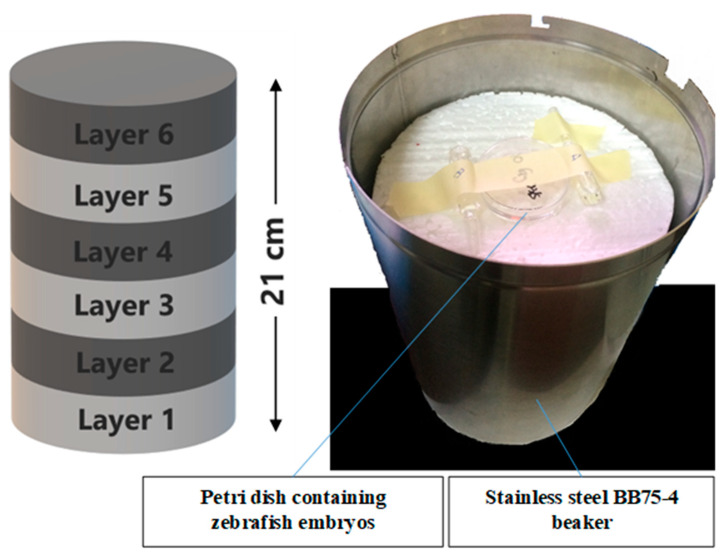
The position of petri dish containing zebrafish embryos in the stainless steel BB75-4 beaker for irradiation.

## Data Availability

Data available in a publicly accessible repository that does not issue DOIs.

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
