# Peer review of "Radioprotective Effects of Kelulut Honey in Zebrafish Model"

_molecules, 2021, doi:10.3390/molecules26061557_

Round 1
Reviewer 1 Report
Dear Author(s)
After an exhaustive revision, the manuscript is RECONSIDER AFTER MAJOR REVISION (CONTROL MISSING IN SOME EXPERIMENTS). In general, the study is closely connected to the journal's objectives. The study is very interesting. However, the manuscript has a lot parts that need to be corrected. The manuscript has many errors in the format, which makes it difficult to read. The introduction is very messy. The results have statistical problems, generating problems when evaluating the novelty/originality of the manuscript. The discussions are very complete.
The manuscript has colloquial English. Therefore, the authors must improve English throughout the manuscript. The authors strongly need to submit the manuscript to an English editor.
In the following pages, I give a detailed revision of the manuscript.
Best regards
GENERAL COMMENTS
Affiliations have certain errors:
- The e-mail hidayat@nm.gov.my (M.N.H.A) is repeated in the affiliation 1 and 2.
- Where are the email and the initials of "Farah Jehan Ahmad Bahri2"?
- The initials must end with ".", i.e., (M.N.H.A.)
- The correspondent author (*correspondence) does not have the necessary data (phone, initials), and in addition, the corresponding email should not be appear in the affiliations.
- All the references in the manuscript were written with surnames, which is wrong, since the format indicates that it must be with numbers. Example: [1]. The authors must see the Author's Guide of “molecules” and follow only one format in all the manuscript. In addition, the manuscript has a lot of disorder in the number of references.
- The manuscript has some problems with the English. Therefore, the authors must improve English throughout the manuscript. The authors need to submit the manuscript to an English editor.
- INTRODUCTION
General comments
The references in the introduction section were written with surnames. This indicates that the authors did not read the author's guide, since the Author's Guide of “molecules” indicates that it must be with numbers. Additionally, the first reference (Mohd Zulmadi et al. (2017)) has the number [36] and the next reference (Adenan et al. (2019)) has the number [3], which is concerning. The authors should start the references from [1] onwards, and furthermore, the reference section should be completely fixed.
It is possible to denote problems in the form of writing. The introduction is very difficult to understand, since it does not present an order from general information to specific information. The section presents disorder in the exposed.
The objective of the study should more clearly.
- Page 2. Line 48. "Foremost among…"
The authors should change these words, since it could confuse or change the meaning of the phrase.
- Page 2. Line 51. “on assembly lines to test consumer goods.”
The authors should change this line, since it could confuse or change the meaning of the phrase.
- Page 2. Line 51. “For an example,”
The authors should change this line to “For example,”
- Page 2. Lines 46-55.
What is(are) the reference(s)?
- Page 2. Line 60. “When radiation…”
The reviewer understands the line. However, the word “when” is common to questions. This word could confuse or change the meaning of the phrase.
- Page 2. Line 66. “on the length of time the individual remains…”
The authors should change this line, since the English is not good.
- Page 2. Lines 67-70.
What is(are) the reference(s)?
- Page 2. Lines 70-71.
These sections need to a connector, since the authors mention the ionizing radiation, and immediately, the authors mention the kelulut bee. What is the connection?
- Page 2. Line 73. “…is reported to have…”
The authors should change this line, since the English is not good.
- Page 2. Lines 77-81.
What is(are) the reference(s)?
- Page 2. Lines 89-97. Page 3. Lines 98-104.
These lines must be reorganized, since previously the authors mentioned honey and without argument, the authors start the lines on ionizing radiation. It is confusing for the reader to follow the introduction of the manuscript.
- Page 2. Lines 77-81. Lines 113-116. Lines 121-123.
What is(are) the reference(s)?
- Page 3. Lines 107-116.
What is the context on Zebrafish? The manuscript is about bee and ionizing radiation.
- Page 2. Lines 121-123.
What is(are) the reference(s)?
- RESULTS
General comments
The authors describe very well the results. However, I have some observations.
- The authors should provide more information on Figure 1 in the text.
- It is not necessary to repeat "p <0.05" and "Figure 2(b)" so many times in the text.
- The Figures 2a shows problems about the statistics. For example, Figure 2a, in IN, 15, 16 and 17 Gy not indicate significant differences. However, according to the authors, 16 and 17 Gy have significant differences 16 with b* and 17 Gy with c*. ¿How can the authors explain that 18, 19 and 20 Gy have significant differences with 15, 16 and 17 Gy?. A similar behavior ocurrs in 17 and 18 Gy with IKH, since the authors indicate significant differences (different letters), but visually the data have no significant differences. The Figure 2b presents the same statistical problems. The Figure 2c presents other problems, in 19 Gy with IKH and IA, since the authors indicate significant difference, but the samples have the same letters (a).
- The authors need to add the range of dimensions in Figures 1, 4, 7 and 9.
- Page 9. I have counted that the authors add the word "p<0.05" 20 times (in a single page), which is completely unnecessary, the authors must be more selective about this word.
- The Figure 5 presents the same statistical problems as mentioned in the Figure 2.
- Page 12. I have counted that the authors add the word "p<0.05" 5 times (in a single page), which is completely unnecessary, the authors must be more selective about this word.
- The Figure 6 presents the same statistical problems as mentioned in the Figure 2 and Figure 5. Additionally, the Figure 6 has a different format than the Figures indicated above.
- Page 14. I have counted that the authors add the word "p<0.05" 4 times (in a single page), which is completely unnecessary, the authors must be more selective about this word.
- The Figure 8 and Figure 10 present the same statistical problems as mentioned in the Figure 2, 5 and 6.
- DISCUSSION
General comments
The discussion should be reflected in a clear and concise explanation of the results obtained, and also, the results should be compared with previous studies.
The authors make an excellent comparison of the results obtained with other studies. However, it is necessary to discuss the differences or equalities with the other studies, i.e., the authors must add more depth to the discussions.
The authors repeat the same reference many times in continuous lines. The authors should only write a reference. Example: Lines 459-464. Lines 515-523.
- Page 18. Lines 429-432. Page 19. Lines 440-445. Lines 453-456.
The lines are similar to a description of results. The authors should improve the lines.
- Section 3.2, 3.3 and 3.4 are well complete with all the details expected in the discussion section
- MATERIALS AND METHODS
General comments
This part is complete and very clear. However, I have some observations:
The authors must add a Figure that represents all the methodology. This Figure will help to understand the methodology in the manuscript.
In this section appears the Figure 1 (page 23). However, in the section Results appears other Figure 1.
In the subsection is not necessary to repeat the references.
Page 23. 4.4 Gamma Irradiation
What is(are) the reference(s)?
Page 24. 4.7 Immunohistochemistry analysis
What is(are) the reference(s)?
- CONCLUSIONS
The English is not good. The authors need to improve the English.
The conclusions need to improve, since this section is incomplete in comparison with all the results in the manuscript.
REFERENCES
- The references have a different format to “molecules”, and this mentions a major oversight by the authors. The authors must see the Author's Guide of “molecules” to the references section.
Author Response
Dear Reviewer,
- I have revised the manuscript and considered all of the comments given. Please find the attachment for the revised manuscript.
Introduction:
- I have corrected and reconstructed the introduction according to the comments.
- Page 2. Line 48. "Foremost among…" - correction Page 2.Line 48-49.
- Page 2. Line 51. “For an example,” - correction Page 2.Line 50-51.
- Page 2. Lines 46-55. - References Page 2. Line 48 - 52.
- Page 2. Line 60. “When radiation…”- I have removed this paragraph.
- Page 2. Line 66. “on the length of time the individual remains…”
- - I have removed this paragraph.
- I have located the suitable references and corrected the sentences according to the comments.
Results:
- According to reviewer comments on the statistical errors in Figure 2a-2c, I want to explain, for the analysis, I have used Duncan's Post Hoc test using SPSS software. For this figure, I try to explain the differences among the treatment groups IN, IKH and IA for each irradiation doses and not the differences among the irradiation doses. This is similar to Figure 5, 6 8(b) and 10(b).
- For Figure 8(a) and 10(b), I have used Dunnet's Post Hoc test using the SPSS software and for these results, I want to explain the differences between the zebrafish in different doses as compared to the negative control.
- For 'p <0.05', I have relocated it only at suitable sentences.
Discussion
- Lines 459-464, Lines 515-523, Page 18. Lines 429-432. Page 19. Lines 440-445. Lines 453-456..- I have rephrased the sentences - Line 419 - 424, 475-483, 395-396, 404-407 and 416-417.
Materials and methods:
- I have relabeled the figure in this section to Figure 11.
- I have inserted the appropriate references according to the comments.
References:
I have revised the references according to the Molecules format
Conclusion:
I have corrected some grammatical errors in the sentences.
Reviewer 2 Report
The paper entitled “Radioprotective Effects of Kelulut Honey in Zebrafish Model” represent interesting data and on my opinion, it falls within the scope of Molecules. This paper is well-written and presents an interesting and carefully designed research. Some minor issues should be resolved before publishing this paper.
In detail:
- Page 4, line 139 - Please explain abbreviation. Full name of "hpf " is required when the abbreviation is used first time.
- The conclusion section is missing completely. The authors should highlight the main achievements of the work in their conclusions. I strongly suggest the authors to try to summarize all the data provided in the manuscript and give a clear idea of the results
- References cited in the text does not meet the requirements of this journal. Same, in the References section, the writing manner of references did not follow the style of this journal. Authors have to check and revise these errors carefully.
In general, the article is valuable and the presented information is novel.
Author Response
Dear reviewer,
- Page 4, line 139 - Please explain abbreviation. Full name of "hpf " is required when the abbreviation is used first time. - "hpf " is hours post fertilization - Line 123.
- I have revised the reference according to the comments given.
- The revised version of the manuscript can be found in the attachment.
Reviewer 3 Report
The work provided by Adenan et. al. aims to investigate the potential of Kelulut honey as a radioprotector. The authors achieve this treating irradiated zebrafish embryos with the honey. Abnormal development and lethality were used as readouts of radioprotection. The authors went further to understand the radioprotective effects at the cellular and molecular level by assessing various markers by IHC. From the experiments, it was concluded that Kelulut honey increases survival and provided organ-specific protection through reducing DNA damage and apoptosis. Kelulut honey may have some added benefit over current radioprotector in that it has less side effects and lower toxicity. Overall, this reviewer found the manuscript difficult to read/follow. It can benefit from a significant amount of revision and more through experimentations.
- IN, IKH,, and IA groups should be defined in the text (line 143)
- Base on currently labeling, it is not clear to this reviewer that is meant by a*, b*, and c* in Figure 2 and 3.
- The death shown is Figure 1 is quite severe and is also observed in embryos after death. For confidence in the assay, the authors should depict several morphologies of the embryos prior to “coagulation”. Figure 4 should be combined with Figure 1 and a more detailed quantification of phenotype distribution should be provided.
- Figure three is unnecessary
- The embryos in Figure 4a appear to be significantly older than 24 hours post fertilization.
- Images from Figure 7 and 9 are not very convincing and could benefit from other readouts such as qPCR or Western Blotting on whole embryos.
Author Response
Dear reviewer,
- I have revised the manuscript and it can be found in the attachment.
- IN, IKH,, and IA groups should be defined in the text (line 143) - correction Line 127
- Base on currently labeling, it is not clear to this reviewer that is meant by a*, b*, and c* in Figure 2 and 3 - I have corrected and can be seen in the attachment.
- The death shown is Figure 1 is quite severe and is also observed in embryos after death. For confidence in the assay, the authors should depict several morphologies of the embryos prior to “coagulation”. Figure 4 should be combined with Figure 1 and a more detailed quantification of phenotype distribution should be provided - For explanation, for this study, the coagulated embryos did not going trough abnormality phase before we found the lethality effect. Only the embryos that survived undergo abnormalities.
- The embryos in Figure 4a appear to be significantly older than 24 hours post fertilization - The embryos for this figure are taken 24 hours post-irradiation which means the age is 48 hours post-fertilization.
Reviewer 4 Report
Authors need to clarify the results and figures better; in particular, figures 2a and 2b must be better explained in fact some comments refer to the panel of figure 2a and others to that of figure 2b in an unclear way (page 4 lines 153-159).
The authors should explain why they analyzed some effects 48 hours after irradiation (Figures 2) and others 72 hours and 96 hours after irradiation (Figures 1 and 4).
The text does not contain the comment of Figure 3.
The authors need to clarify the effects of the different doses (11Gy to 20Gy).
Did the authors not analyze the phenolic content of honey?
Throughout the manuscript, the bibliography does not respect the actual numbering (eg Page 2 line 55, reference 36 must be corrected with 1, and so on).
on page 12, line 318 (72) should be corrected with (72h).
In the manuscript the hours have to be standardized (h or hours) (page 12 lines 329- 336).
On page 3 line 102 and on page 21 line 545 in vitro should be written in italics (in vitro).
On page 22, line 588 in vivo should be written in italics (in vivo).
Author Response
Dear reviewer,
- I have revised the manuscript and it can be found in the attachment.
- The authors should explain why they analyzed some effects 48 hours after irradiation (Figures 2) and others 72 hours and 96 hours after irradiation (Figures 1 and 4).- The effects according to OECD guidelines must be observed for 96 hours after treatment.
- The text does not contain the comment of Figure 3.-I have corrected the Figure 2(c) to 3.
- The authors need to clarify the effects of the different doses (11Gy to 20Gy).- This already clarified in the discussion.
- Did the authors not analyze the phenolic content of honey?-The phenolic contents of the KH (stingless bee honey) has been published in many literatures. We just mentioned the effectiveness of the honey since our focus is more on radioprotective effects.
- Throughout the manuscript, the bibliography does not respect the actual numbering (eg Page 2 line 55, reference 36 must be corrected with 1, and so on).- I have revised the all of the references.
- Hours has been standardized as "h"
- On page 3 line 102 and on page 21 line 545 in vitro should be written in italics (in vitro).- I have corrected the sentence.
On page 22, line 588 in vivo should be written in italics (in vivo). - I have corrected the sentence.
Round 2
Reviewer 1 Report
Dear Author(s)
After an exhaustive revision, the manuscript is Accept in present form. The resubmitted manuscript has been completely improved compared to its previous version. Therefore, the manuscript can be published in “Molecules”.
Best regards
Reviewer 3 Report
Most concerns of this reviewer have been addressed.